

# Population genetics and demography of the coral-killing cyanobacteriosponge, *Terpios hoshinota,* in the Indo-West Pacific

Savanna Wenhua Chow[1,2,3,4,*], Shashank Keshavmurthy[2,*], James Davis Reimer[5,6], Nicole de Voogd[7,8], Hui Huang[9], Jih-Terng Wang[3], Sen-Lin Tang[2], Peter J. Schupp[10,11], Chun Hong Tan[12], Hock-Chark Liew[13], Keryea Soong[3], Beginer Subhan[14], Hawis Madduppa[14] and Chaolun Allen Chen[1,2,4,15]

[1] Department of Life Sciences, National Taiwan Normal University, Taipei, Taiwan
[2] Biodiversity Research Center, Academia Sinica, Taipei, Taiwan
[3] Department of Oceanography, National Sun Yat-sen University, Kaohsiung, Taiwan
[4] Biodiversity Program, Taiwan International Graduate Program, Academia Sinica, Taipei, Taiwan
[5] Department of Biology, Chemistry, and Marine Science, University of Ryukyus, Naha, Okinawa, Japan
[6] Tropical Biosphere Research Center, University of the Ryukyus, Okinawa, Japan
[7] Naturalis Biodiversity Center, Leiden, The Netherlands
[8] Institute of Environmental Sciences, Environmental Biology Department, Leiden University, Leiden, Netherlands
[9] CAS Key Laboratory of Tropical Marine Bio-resources and Ecology and Guangdong Provincial Key Laboratory of Applied Marine Biology, South China Sea Institute of Oceanology, Chinese Academy of Sciences, Guangzhou, China
[10] Institute for Chemistry and Biology of the Marine Environment, University of Oldenburg, Oldenburg, Germany
[11] Helmholtz Institute for Functional Marine Biodiversity at the, University of Oldenburg (HIFMB), Oldenburg, Germany
[12] School of Marine and Environmental Sceinces, University of Malaysia Terengganu, Terengganu, Malaysia
[13] Sdn Bhd. Jalan Hiliran, Kuala Terengganu, Alchemy Laboratory & Services, Terengganu, Malaysia
[14] Department of Marine Science & Technology, Faculty of Fisheries & Marine Sciences, IPB University, Bogor, Indonesia
[15] Department of Life Science, Tunghai University, Taichung, Taiwan
[*] These authors contributed equally to this work.

Corresponding authors
Hawis Madduppa,
madduppa@gmail.com,
hawis@apps.ipb.ac.id
Chaolun Allen Chen,
cac@gate.sinica.edu.tw

## ABSTRACT

The first occurrence of the cyanobacteriosponge *Terpios hoshinota* was reported from coral reefs in Guam in 1973, but was only formally described in 1993. Since then, the invasive behavior of this encrusting, coral-killing sponge has been observed in many coral reefs in the West Pacific. From 2015, its occurrence has expanded westward to the Indian Ocean. Although many studies have investigated the morphology, ecology, and symbiotic cyanobacteria of this sponge, little is known of its population genetics and demography. In this study, a mitochondrial cytochrome oxidase I (COI) fragment and nuclear ribosomal internal transcribed spacer 2 (ITS2) were sequenced to reveal the genetic variation of *T. hoshinota* collected from 11 marine ecoregions throughout the Indo-West Pacific. Both of the statistical parsimony networks based on the COI and nuclear ITS2 were dominated by a common haplotype. Pairwise $F_{ST}$ and Isolation-by-distance by Mantel test of ITS2 showed moderate gene flow existed among most populations in the marine ecoregions of West Pacific, Coral Triangle, and Eastern

Indian Ocean, but with a restricted gene flow between these regions and Maldives in the Central Indian Ocean. Demographic analyses of most *T. hoshinota* populations were consistent with the mutation-drift equilibrium, except for the Sulawesi Sea and Maldives, which showed bottlenecks following recent expansion. Our results suggest that while long-range dispersal might explain the capability of *T. hoshinota* to spread in the IWP, stable population demography might account for the long-term persistence of *T. hoshinota* outbreaks on local reefs.

# INTRODUCTION

Competition among sessile organisms acts as one of the ecological factors for shaping the biodiversity and community structure of coral reefs (reviewed in *Norström et al., 2009*; *Chadwick & Morrow, 2011*). As anthropogenic disturbances such as global climate change, overfishing, pollution, coral disease, and invasive species increase (*Hughes et al., 2003*; *Hoegh-Guldberg et al., 2007*), these threats may weaken zooxanthellate reef-building corals in their encounters with other sessile organisms during competitive interactions. This eventually leads to the alteration of benthic organism composition from coral dominance to communities dominated by competitors such as macroalgae, sponges, ascidians, zoantharians, sea anemones, or corallimorpharians (*Hatcher, 1984*; *Chen & Dai, 2004*; *Loya, 2004*; *Norström et al., 2009*; *Reimer et al., 2021*).

Cyanobacteriosponges associated with photosynthetic cyanobacteria comprise one benthic group that competes for substrates with reef-building corals due to fast growth (faster than sponges without photosynthetic symbionts) and aggressiveness, resulting in the overgrowth and killing of corals (*Vicente, 1990*; *Rützler & Muzik, 1993*; *Diaz' et al., 2007*, reviewed in *Usher, 2008*). *Terpios hoshinota* (*Rützler & Muzik, 1993*) (Family Subertidae) is an encrusting cyanobacteriosponge with a growth rate of approximately 1−2.5 mm/day (*Plucer-Rosario, 1987*; *Rützler & Muzik, 1993*; *Soong, Yang & Allen Chen, 2009*). Moreover, due to its flat (*Wilkinson, 1983*; *Wilkinson, 1987*), thin, sheet-like (<1 mm) structure, *T. hoshinota* firmly encrusts corals by penetrating deeply into the coral skeleton (*Hirose & Murakami, 2011*, see also *Wang et al., 2015*). Another characteristic of this sponge is the presence of extremely dense populations of photosynthetic cyanobacteria in the mesohyl (*Bryan, 1973*; *Plucer-Rosario, 1987*; *Rützler & Muzik, 1993*; *Liao et al., 2007*; reviewed in *Usher, 2008*). These cyanobacteria belong to a group closely related to *Prochloron* spp., (occupy >50% of total cell density, and contribute to the species' gray/black coloration (*Rützler & Muzik, 1993*; *Hirose & Murakami, 2011*; *Tang et al., 2011*). The efficiency of *T. hoshinota* to grow on living corals rather than other substrates (*Chen, Kuo & Chen, 2009*) is because of its plastic morphological features, including hairy tips packed with cyanobacteria, sponge tissues, and spicules (*Wang et al., 2012a*), and the ability to transform from an encrusting sheet-like structure to a thread-like tissue in order to move

across a shaded area or reach new territories (*Soong, Yang & Allen Chen, 2009*; *Wang et al., 2012a*). *Terpios hoshinota* often outcompetes coral competitors and alters ecosystem function (*Madduppa et al., 2017*; *Nugraha, Zamani & Madduppa, 2020*). Occasionally, the species exhibits negative growth and can even be overgrown by certain coral species or red calcareous algae (*Plucer-Rosario, 1987*; *Wang et al., 2012a*). Photo-physiological measurements and stable isotope analyses have shown that *T. hoshinota* competes with corals not only by morphological transformation of the sponge-cyanobacteria association, but also by physiologically outperforming opponents in accumulating resources for competition (*Wang et al., 2015*).

These unique characteristics give *T. hoshinota* a 'coral-killing' capacity and allow it to quickly overgrow coral colonies or other hard substrates. Consequently, the species has caused significant degradation in coral communities at many reefs across the Indo-West Pacific (IWP) (*Marine Park Center Foundation, 1986*; *Rützler & Muzik, 1993*; *Liao et al., 2007*; *Chen, Kuo & Chen, 2009*; *Reimer et al., 2011*; *Shi et al., 2012*). The occurrence of *T. hoshinota* was first reported on the fringing reefs of Guam in 1973 (*Bryan, 1973*). Among 37 reefs surveyed in Guam, 25 had almost-completely encrusted coverages of *T. hoshinota* that varied in size from a few small patches to areas up to 1,000 $m^2$ (*Bryan, 1973*; *Plucer-Rosario, 1987*), and in some reef sites *T. hoshinota* still persists to the present day (*Myers & Raymundo, 2009*; *Schils, 2012*).

In the last four decades, *T. hoshinota*- associated rapid declines in coral cover have been reported from reefs in Okinawa, Japan (*Marine Park Center Foundation, 1986*; *Rützler & Muzik, 1993*; *Reimer, Nozawa & Hirose, 2010*; *Reimer et al., 2011*; *Yomogida et al., 2017*), and Green and Orchid Islands, Taiwan, causing significant declines in living coral cover (*Liao et al., 2007*; *Chen, Kuo & Chen, 2009*; *Soong, Yang & Allen Chen, 2009*). In addition to the suggested role of *T. hoshinota* in the degradation of the living coral cover and their decline between 2002 and 2008 at the Yongxing Island reef, China (*Shi et al., 2012*), the species has also been recorded from Lizard Island in the Great Barrier Reef, Australia (*Fujii et al., 2011*), Tioman Island, Malaysia (*Hoeksema, Waheed & de Voogd, 2014*), and the Thousand Islands (NW Java) and Spermonde Archipelago (SW Sulawesi), Indonesia (*DeVoogd, Cleary & Dekker, 2013*; *Hoeksema, Waheed & de Voogd, 2014*; *vander Ent, Hoeksema & de Voogd, 2016*; *Madduppa et al., 2017*; *Utami, Zamani & Madduppa, 2018*; *Nugraha, Zamani & Madduppa, 2020*). Recent observations also confirmed the occurrence of *T. hoshinota* in the Indian Ocean, including the Maldives (*Montano et al., 2014*), Mauritius (*Elliott et al., 2016a*; *Elliott et al., 2016b*), and Kimberley inshore reefs, Western Australia (*Fromont, Richards & Wilson, 2019*). This biogeographic information suggests a possible widespread distribution of *T. hoshinota* in the IWP and makes the species a potential widespread threat to corals and coral reefs (Table S1).

While many studies have looked at the mechanisms of *T. hoshinota*'s competitive success, little is known about its population structure and demographic history based on molecular genetics across its range. Most studies on *T. hoshinota* have focused on its ecology, physiology (*Soong, Yang & Allen Chen, 2009*; *Hirose & Murakami, 2011*; *Wang et al., 2012a*; *Wang et al., 2012b*; *DeVoogd, Cleary & Dekker, 2013*; *Wang et al., 2015*; *Nozawa, Huang & Hirose, 2016*; *Nugraha, Zamani & Madduppa, 2020*), distribution (*Elliott et al.,*

*2016a*; *Elliott et al., 2016b*; *Madduppa et al., 2017*; *Yang et al., 2018*; *Diraviya Raj et al., 2018*; *Ashok et al., 2019*; *Das et al., 2019*; *Thinesh et al., 2015*), habitat preferences (*Schils, 2012*; *Hsu, Wang & Chen, 2013*; *vander Ent, Hoeksema & de Voogd, 2016*; *Elliott et al., 2016a*; *Elliott et al., 2016b*), and interactions between the sponge and other benthic organisms (*Reimer et al., 2011*; *Wang et al., 2012a*; *Wang et al., 2012b*; *Hoeksema, Waheed & de Voogd, 2014*; *Thinesh et al., 2017a*; *Thinesh et al., 2017b*; *Syue, Hsu & Soong, 2021*). *vander Ent, Hoeksema & de Voogd (2016)* examined the genetic variation of *T. hoshinota* found within the Spermonde Archipelago, Indonesia, by sequencing partitions of the mitochondrial COI (COI) and nuclear ribosomal 28S gene (28S). Two COI haplotypes that differed from the COI sequence in GenBank were discovered within the archipelago. Recent surveys on coral reef biodiversity revealed the first report of *T. hoshinota* in Kimberley inshore reefs, Western Australia (*Fromont, Richards & Wilson, 2019*) suggesting that *T. hoshinota* may have arrived recently in the Kimberley, possibly after 2014 (*Fromont & Sampey, 2014*; *Bryce, Bryce & Radford, 2018*), although it may have been there previously but not detected. Interestingly, *T. hoshinota* in the Kimberley shared an identical COI haplotype with its counterpart from the Spermonde Archipelago, Indonesia, which might imply that there is trans-equatorial dispersal in this sponge. To better understand how *T. hoshinota* disperses and outbreaks, broad-scale sampling and examination of genetic variation throughout its biogeographic range is needed.

In this study, we sampled *T. hoshinota* from its currently known distribution in the IWP and examined the variation in its COI and nuclear ribosomal internal transcribe spacer (ITS) sequences. By revealing population genetic structure and demographic history, potential hypotheses of how *T. hoshinota* disperses and colonizes in the IWP are proposed.

## MATERIAL AND METHODS

### Sampling

All specimens were collected with permission and sampling permits where required (permits not required for sampling in Japan, Taiwan, China). A total of 234 specimens of *Terpios hoshinota* were examined in this study (Fig. 1, Table 1). We collected specimens at each location ranging from a minimum of 1 to a maximum of 41 specimens. This variation and non-uniformity in sampling was due to several factors: (1) variation in numbers of *T. hoshinota* at different sites, (2) field workers not identifying *T. hoshinota* in the field, and (3) sampling restrictions in some cases (*e.g.*, the Great Barrier Reef). After collection, macro-invertebrates and commensals were carefully removed. All samples were preserved in 75% ethanol inside centrifuge tubes and stored at −20 °C until extraction.

For Guam, samples were collected with permission of the Department of Agriculture Division of Aquatic and Wildlife Resources (DAWR) Guam and certificate of origin from DAWR was obtained prior to shipping. Permit is not required to collect sponges in Green Island and in China. For Indonesia, permit was issued by the Indonesian Ministry of Research, Technology & Higher Education (KLN RISTEKDIKTI scheme, 2016-2018) No. 011/SP2H/LT/DRPM/IV/2017. For Malaysia, sampling permit was issued by the Marine Park and Resource Management Division, Department of Fisheries Malaysia (formerly

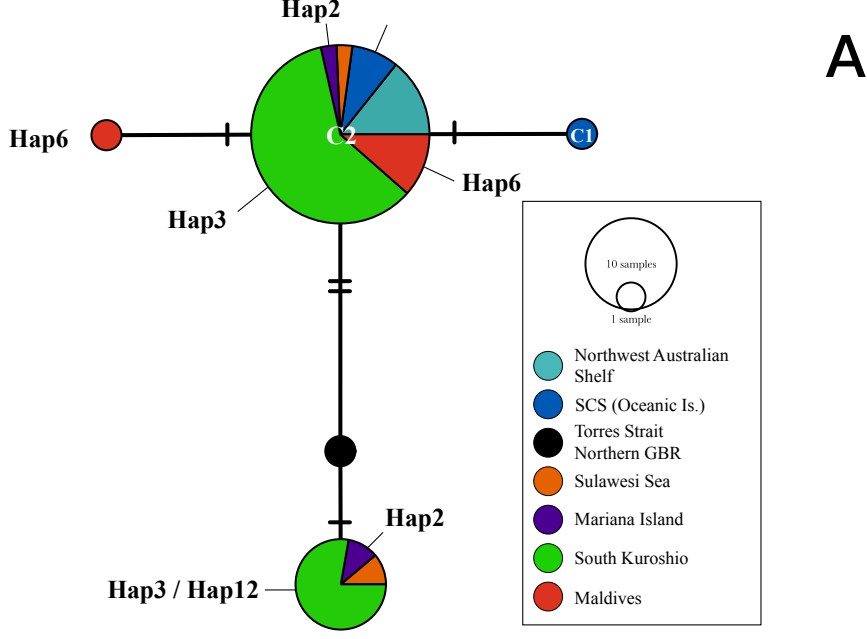

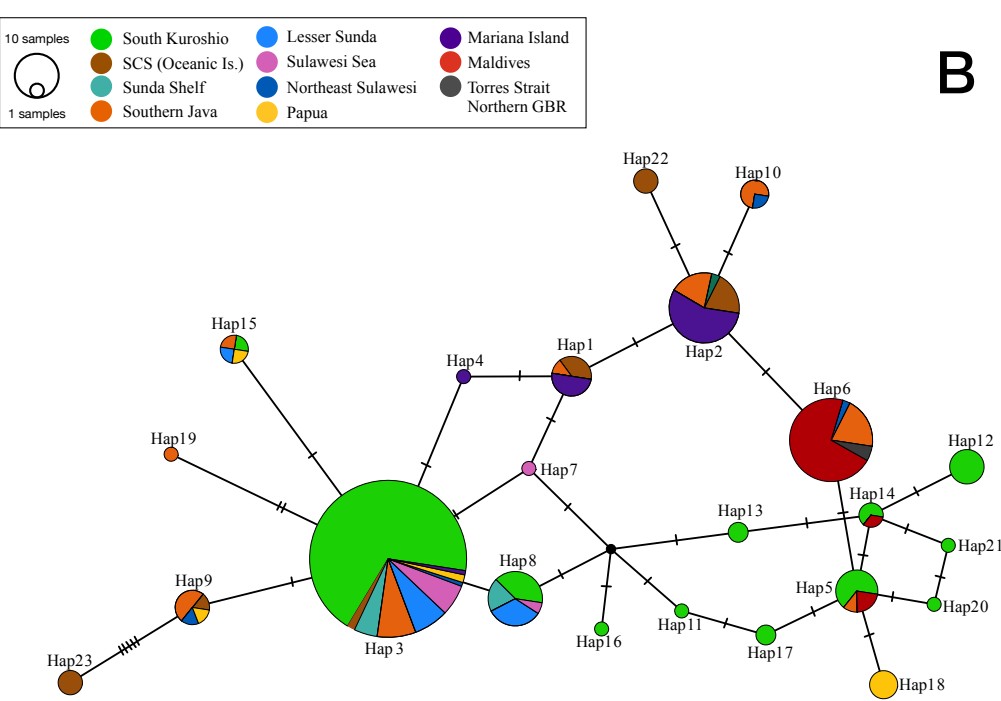

**Figure 1** **Statistical parsimony network showing relationships among (A) cytochrome oxidase I and (B) internal transcribed spacer 2 sequences from *T. hoshinota* population based on marine ecoregion.** The circle size is relative to the number of haplotypes present in the dataset, small black circles indicating nodes, hatch marks indicating missing haplotypes, and "Hap" represents for Haplotypes in both ITS and COI markers. Haplotypes are colored by the definition of marine ecoregion.

Department of Marine Park Malaysia) with a Sampling permit: JTLM 630-7 JLD. 4(8). Samples from Australia were part of Census of Marine Life/Census of Coral Reefs, with permit No. G32313.1 issued by the Great Barrier Reef Marine Park Authority.

## DNA extraction and PCR amplification

Genomic DNA was extracted from 25 mg of clean sponge tissue using the DNAeasy-Tissue Kit (Qiagen) according to the manufacturer's instructions. Specimens used for COI sequencing were from Green Island ($n = 12$), Lanyu (3), Wanlitung (1), Guam (2), Maldives (5), Miyako (2), Bise (6), Shiraho (1), Yakomo (2), Lizard Island (1), and Xisha (3). A fragment containing 650-bp COI of *T. hoshinota* was amplified by the universal primer set (*Folmer et al., 1994*) LCO 1490: 5′-GGTCAACAAATCATAAAGATATTGG-3′) and HCO 2198: 5′-TAAACTTCAGGGTGACCAAAAAATCA-3′). The thermal cycles for PCR consisted of 1 cycle at 94 °C for 2 min, followed by 35 cycles at 94 °C for 50 s, 40 °C for 55 s, 72 °C for 60 s, and a final cycle of 72 °C for 7 min. The amplified products were directly sequenced in both directions (Mission Biotech Taiwan, http://www.missionbio.com.tw).

In addition, universal primers for amplifying the entire nuclear ribosomal internal transcribed spacer (ITS), (ITS4: 5′-TCCTCCGCTTATTGATATGC-3′; EF3RCNL: 5′-CAAACTTGGTCATTTAGAGGA-3′) (*Lord et al., 2002*), were used to design specific ITS 2 primers for *T. hoshinota* as below. The number of samples used for cloning were; Green Island ($n = 9$) LiuMaGou (2), ChaiKou (1), BiTouXi (2), ZhaoRi hot spring (2), YouZi Lake (1) and Shiraho, Japan (1). DNA samples were amplified by PCR with 1 cycle at 94 °C for 3 min, 94 °C for 30 s, 54 °C for 90 s, and 72 °C for 60 s at 35 cycles, followed by a final cycle of 72 °C for 10 min. The length of the PCR products (∼650 bp) was then confirmed by gel electrophoresis (1% agarose). PCR products were then cloned by using an Invitrogen TA cloning kit (Thermo Fisher Scientific, https://www.thermofisher.com/tw/zt/home/life-science/cloning/ta-cloning-kits.html) following the manufacturers protocol. Treated plasmids containing Ampicillin (50 μL), IPTG (0.5 mM), X-Gal (50 μg/ml) were then added into Luria Bertani agar plates for cloning. Positive clones were selected during the transformation step and a minimum of 20 clones were picked up for each set of sponge samples, followed by PCR with universal primer (ITS4, EF3RCNL). A total of 111 positive clones were selected and amplified for the completed ITS region. Cloned ITS DNA sequences were submitted to GenBank using the *blastn* option, and sequences that matched 100% to non-poriferan species were excluded from the following analyses. Of the 111 ITS clones, 72 sequences matched sponge species (*Hymeniacidon heliophila*, Accession: AB250764) with 80 to 89% similarity values (Fig. S2). Primers for highly variant regions of internal transcribed spacer 2 (ITS2) were designed by Geneious Pro 4.5 "Primer3" (*Untergasser et al., 2012*) module with default value and rectified its structure to prevent hairpin structures or self-annealing problem by Primer Select (Lasergene). The designed primer set was ITS2Fwd: 5′-TTTGCAGACGGACAGCCTCA -3′, ITS2Rev: 5′- TTTTACTGTGCACCCCTCTCAG-3′). Temperature Gradient PCR test was performed to find the best annealing temperature for ITS2 Primer, temperature starting from 47 °C to 65 °C were set as temperature gradient scope coming by 10 steps of temperature gradient with each gradient steps up at 2 °C. Gel

**Table 1** **Genetic diversity index and neutrality tests of *T. hoshinota*.** Ecoregion, Sample size ($n$), phased sequence number (seqs), haplotypes ($h$), nucleotide diversity ($\pi$), haplotype diversity ($h_d$), p-distance(p-dist), standard deviation(SD), Tajima's D, Fu's FS, mismatch distribution, Ramos-Onsins' R2 including associated $p$-values and the current expansion status inferred from the neutrality test.

| Ecoregion | n | seqs | h | π (SD) | $h_d$ (SD) | p-dist (SD) | Tajima's D | Fu's $F_S$ | Mismatch distribution | Ramos-Onsins' $R^2$ | Expansion |
|---|---|---|---|---|---|---|---|---|---|---|---|
| South Kuroshio | 109 | 115 | 12 | 0.0107 (0.00160) | 0.435 (0.058) | 1.373 (0.562) | 0.0998 | −3.959 | Multimodal | 0.0696 | No |
| South China Sea Ocean Islands | 9 | 17 | 6 | 0.0245 (0.00490) | 0.853 (0.047) | 3.132 (1.064) | 1.1448 | 0.671 | Bimodal | 0.1958 | No |
| Sunda Shelf | 10 | 10 | 3 | 0.0083 (0.00357) | 0.600 (0.131) | 1.067 (0.558) | −0.943 | 0.603 | Biomdal | 0.2324 | No |
| Southern Java | 23 | 32 | 9 | 0.0176 (0.00120) | 0.835 (0.039) | 2.246 (0.947) | 0.8501 | −1.586 | Multimodal | 0.1594 | No |
| Lesser Sunda | 14 | 15 | 3 | 0.0046 (0.00800) | 0.562 (0.095) | 0.610 (0.484) | −0.0238 | −0.064 | Unimodal | 0.1724 | No |
| Sulawesi Sea | 9 | 10 | 2 | 0.0031 (0.00161) | 0.378 (0.181) | 0.400 (0.274) | −1.4009 | −1.164[*] | Unimodal (skewed) | 0.2 | Yes |
| Northeast Sulawesi | 3 | 4 | 4 | 0.0221 (0.00509) | 1.000 (0.177) | 2.833 (1.224) | 0.3719 | −1.322 | Biomdal | 0.159 | No |
| Papua | 6 | 8 | 4 | 0.0173 (0.00267) | 0.750 (0.139) | 2.214 (1.050) | 1.8739 | 0.96 | Biomdal | 0.2768 | No |
| Mariana Islands | 24 | 36 | 5 | 0.0104 (0.00116) | 0.625 (0.065) | 1.333 (0.735) | 1.911 | 0.581 | Bimodal | 0.2222 | No |
| Maldives | 25 | 28 | 3 | 0.0021 (0.00109) | 0.204 (0.098) | 0.269 (0.214) | −1.164[*] | −1.828[*] | Unimodal (skewed) | 0.1052 | Yes |
| Torres Strait Northern GBR | 2 | 2 | 1 | 0 | 0 | 0 | – | – | – | – | – |
| Total/mean | 234 | 277 | 23 | 0.0174 (0.00079) | 0.736 (0.025) | 2.232 (0.863) | −0.337 | −8.081[*] | Bimodal | 0.0582 | |

**Notes.**

n, Sample size; seqs, phased sequence number; $h$, haplotypes; $\pi$, nucleotide diversity; $h_d$, haplotype diversity; p-dist, p-distance; SD, standard deviation.

[*] $P < 0.05$.

electrophoresis images showed that the best annealing temperature was 53 °C. Optimum PCR condition of ITS2 primers are: 1 Cycle at 94 °C for 3 min, 94 °C for 30 s with 53 °C for 30 s and 72 °C for 30 s at 35 cycles, then finished by 1 cycle of 72 °C for 10 min. These new primers were then used for subsequent PCR amplification and direct sequencing of the remaining *T. hoshinota* specimens (Genomics Taiwan, https://en.genomics.com.tw).

## Data analyses

Due to unequal sampling numbers across different reefs, we collapsed specimen groupings (Table 1, Tables S2, S3) based on marine ecoregions (*Spalding et al., 2007*) to obtain better resolution on the genetic diversity of *T. hoshinota* in the IWP. DNA sequences were cleaned visually based on quality and accuracy using Geneious R10 (https://www.geneious.com). Sequences were trimmed and aligned by the MAFFT algorithm (*Katoh & Standley, 2013*) for downstream analyses. All COI sequences obtained in this study were submitted to GenBank with accession numbers MZ914515–MZ914532 and OK576574–OK576593. Previously reported COI haplotypes available from GenBank were downloaded and added to our dataset (KJ008098, MN507872, MN507873, KP764915, KP764916, MN507874–MN507878). Genetic diversity indices nucleotide diversity ($\pi$), number of haplotypes ($h$), and haplotype diversity ($h_d$) were calculated using DnaSP V.6.12.03 (*Rozas et al., 2017*). Mean uncorrected pairwise genetic distances (*p*- distance) were calculated with MEGA 7 (*Kumar, Stecher & Tamura, 2015*). Relationships among sequenced samples were analyzed using a statistical parsimony network algorithm implemented in PopART V.1.7 (*Leigh & Bryant, 2015*).

ITS2 chromatograms were analyzed following the procedure for nuclear sequence markers described in (*Fontaneto, Flot & Tang, 2015*; *Fontaneto, Flot & Tang, 2015*). Sequences were phased using the statistical website phylogeny.fr. Phasing was trivial for homozygous sponges (that had no double peaks) and sponges that had only one double peak (*i.e.*, comprising two ITS1 sequence types that differed at a single position). Sponges that had only a few double peaks, as expected from the superposition of two ITS2 sequence types of equal lengths differing at several positions, were phased using the programs PHASE V.2.1.1 (*Stephens, Smith & Donnelly, 2001*) and SeqPHASE (*Flot, 2010*) where the allele combinations with probability scores above 0.8 were chosen for the analyses. Using the ITS2 marker, 234 samples of *T. hoshinota* were successfully sequenced. There were 191 homozygous and 43 heterozygous individuals, and hence a total of 277 sequences corresponding to 23 haplotypes (128 bp long) was obtained (Table 1). All sequences were submitted to GenBank with accession numbers MZ468904–MZ469137. Haplotype networks were constructed with a statistical parsimony network algorithm (*Templeton, Crandall & Sing, 1992*) by PopART V.1.7 software based on marine ecoregions (*Spalding et al., 2007*). Genetic diversity indices mean uncorrected pairwise genetic distances (*p*- distance) were calculated as described in the previous section (COI sequence analysis). Best-fit models for nucleotide substitution were performed by jModelTest V.2.1.10 (*Darriba et al., 2020*), with the most appropriate evolutionary model TrN for the nucleotide substitution model of the ITS2 sequences. To test past demographic events, we used Tajima's D (*Tajima, 1989*) and Fu's Fs (*Fu, 1997*) in DnaSP

with 10,000 simulations under the selective neutrality model. Mismatch distribution graphs were plotted in DnaSP V.6.12.03 to reveal evidence of spatial range expansion or stable population (*Tajima, 1989*). Demographic changes were also examined by the Ramos-Onsins' $R_2$ in DnaSP V.6.12.03 with 10,000 permutations, which is more powerful for detecting past demographic events than Harpending's raggedness index (*Harpending et al., 1993*) or Fu's Fs when sample size is relatively small (*Ramos-Onsins & Rozas, 2002*). $R_2$ can detect recent strong population growth, and a non-significant value shows no deviation to the null hypothesis and thus represents a constant population size (*Ramos-Onsins & Rozas, 2002*). Population differentiation was examined using the Weir and Cockerham $F_{ST}$ estimator (*Weir & Cockerham, 1984*) by Arlequin 3.5 (*Excoffier & Lischer, 2010*), and significance level ($p = 0.001$) was assessed using 10,000 permutation replicates. Global $F_{ST}$ and average $F_{ST}$ per population following with UPGMA clustering of population were obtained from GraphPad Prism 9. Isolation-by-distance analysis (*Slatkin, 1987*; *Slatkin, 1993*) was performed through linearized pairwise $F_{ST}$ estimates ($F_{ST}/1 - F_{ST}$), and correlated geographical distances of *T. hoshinota* populations based on marine ecoregions using the standard Mantel test implemented in GENODIVE V.3.05 (*Meirmans, 2020*). The linear regression of linearized pairwise $F_{ST}$ values and log transformed geographical distances were plotted by GraphPad Prism 9.

## RESULTS

### Mitochondrial COI genetic diversity and statistical parsimony network
The COI sequences obtained from 37 *Terpios hoshinota* specimens across 10 populations in the IWP were nearly identical, with only 3 phylogenetically informative sites identified. By adding 10 sequences from GenBank, the COI sequences for *T. hoshinota* formed a single statistical parsimony network comprising five haplotypes, with 1 to 2 mutational steps evident among sequences (Fig. 1A) having a mean nucleotide diversity ($\pi$) of 0.00185, haplotypic diversity ($h_d$) of 0.416, and *p*-distance of 1.09 (Table S3), respectively. C2 was the dominant haplotype ($n = 35$, 74.47%) in specimens from the major marine ecoregions, except for specimens from the Torres Strait (Northern GBR). C3 was ranked as the second-most dominant haplotype ($n = 9$, 14.89%) in samples from Guam, Indonesia, and southern Kuroshio. The remaining three COI haplotypes were each identified from single individual specimens from Xisha, the GBR, and the Maldives. Due to the small sample sizes and relatively low genetic variability, COI was not utilized in the historical demography and population genetic analyses.

### ITS2 genetic diversity, demography, population differentiation, and statistical parsimony network
The number of samples (n), nucleotide diversity ($\pi$), *p*-distance, haplotype (*h*), and haplotype diversity ($h_d$) of ITS2 in *Terpios hoshinota* based on marine ecoregions are shown in Table 1. Identical ITS2 sequences in two samples from the Torres Strait, northern GBR, were excluded from the comparisons of $\pi$, $h_d$, *p*- distance, and following demography and $F_{ST}$ -statistics. The mean *p*-distances ranged from 0.269 in the Maldives to 3.132 in the South China Sea Oceanic Islands. Similar patterns were seen in nucleotide diversity, with

0.0021 in the Maldives and 0.00245 in the South China Sea Oceanic Islands. The number of haplotypes ranged from 1 in the Torres Strait, northern GBR, to 12 in South Kuroshio, with haplotype diversity ranging from 0.117 in the Maldives to 1 in Northeast Sulawesi.

The historical demography of *Terpios hoshinota* inferred by Tajima's D was positive for most populations except those from Sunda Shelf, Sulawesi Sea, and the Maldives (Table 1). The results of Fu's Fs test also showed positive values for most populations except those from South Kuroshio, Southern Java, Sulawesi Sea, northeast Sulawesi, and the Maldives. Only the Sulawesi Sea and Maldives had both negative and statistically significant Tajima's D and Fu's Fs, indicating an excess of rare haplotypes over what would be expected under neutrality and rejecting the null hypothesis of constant population size at mutation-drift equilibrium; *i.e.,* Sulawesi Sea and Maldives populations might be experiencing population expansions or be under positive selection. The remaining populations were not statistically diverged from neutrality, suggesting their population sizes were at mutation-drift equilibrium and none significantly deviated from neutrality (Ramos-Onsins' $R_2$, Table 1). Four types of mismatch distribution models were observed in *Terpios hoshinota* across the IWP: unimodal, skewed unimodal, multimodal, and bimodal (Table 1, Fig. S1). While Lesser Sunda showed a unimodal distribution that suggested an association with a sudden expansion model, both the Sulawesi Sea and the Maldives showed skewed unimodal distributions that suggested recent expansions or bottleneck events. The remaining populations had either multimodal or bimodal distributions that suggested constant population sizes or the presence of secondary contacts of two distinct lineages (*Alvarado Bremer et al., 2005*; *Jenkins, Castilho & Stevens, 2018*).

Two hundred and seventy-seven ITS2 sequences of *T. hoshinota* were categorized into 23 haplotypes across the IWP (Fig. 2), forming a single statistical parsimony network comprising 1 to 5 mutational steps evident among specimens (Fig. 1B). H3 was the most dominant haplotype ($n = 133$, 48.01%) present in all sampling sites except the Maldives, GBR, Xisha, and Yakomo (Fig. 2). H6 was the second-most dominant haplotype ($n = 35$, 12.64%), and present only in sites near the equator or in the southern hemisphere, including the Sulawesi Sea, Southern Java, the Maldives, and GBR. H2 was the third-most dominant haplotype ($n = 31$, 11.91%), present in samples from Guam, South China Sea Oceanic Islands, Sunda Shelf, and Southern Java. The rest of the haplotypes were present either in low frequencies or restricted to single populations; for example, H22 and H23 occurred in Xisha, China, H19 in southern Java, and H18 in Papua, Indonesia (Figs. 1B, 2).

Population differentiation based on marine ecoregions was examined by pairwise $F_{ST}$-statistics (Fig. 3, Table S4). Eighteen of 55 (32.73%) pairwise $F_{ST}$ values were significantly different ($p < 0.001$), with negative $F_{ST}$ values in South Kuroshio - Sulawesi Sea ($-0.003$), South Java - Northeast Sulawesi ($-0.1017$), and Maldives –GBR ($-0.286$). Negative $F_{ST}$ values indicate more variation within than between populations or results from uneven sample sizes (*Holsinger & Weir, 2009*; *Gerlach et al., 2010*). The $F_{ST}$ heatmap (Fig. 3) clearly showed that the Maldives, located in the central Indian Ocean, had significant pairwise $F_{ST}$ values (except one with Northeast Sulawesi) in comparison with all other populations in the marine ecoregions of the eastern Indian Ocean, Coral Triangle, and West Pacific, with the largest $F_{ST}$ (0.6687) shown between the Maldives and Sunda Shelf. Testing for

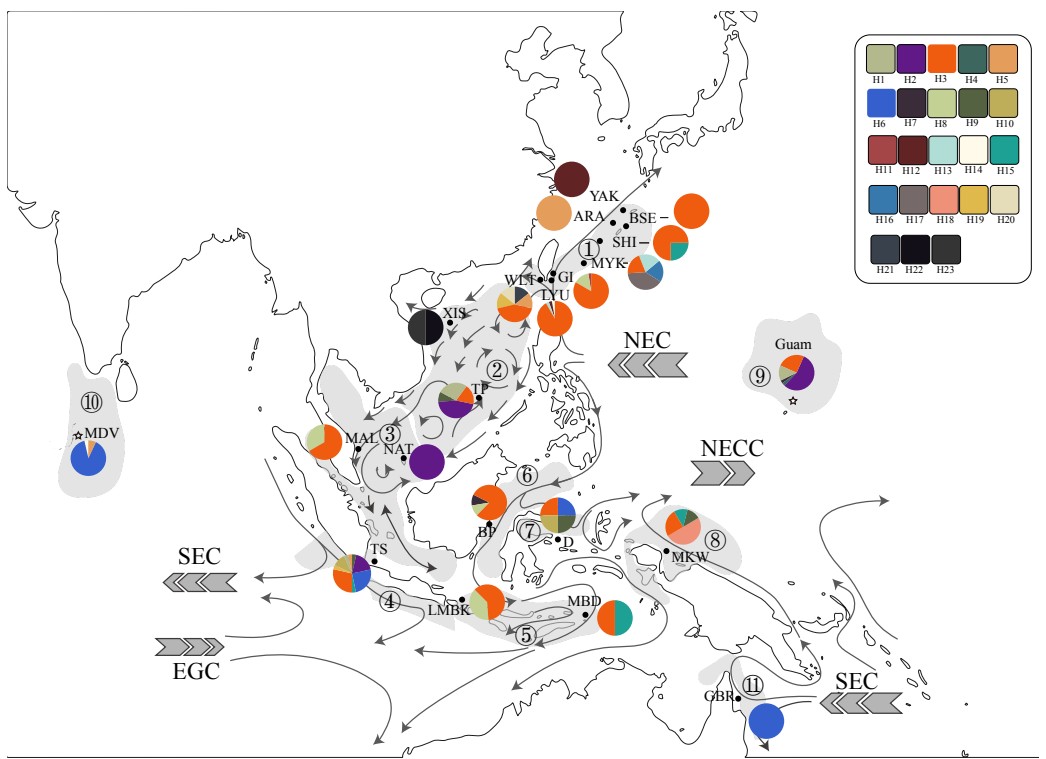

**Figure 2  Haplotype map of ITS2 sequences of 234 *T. hoshinota* specimens from 22 sampling sites with geographical range of marine ecoregion in this study.** The alignment comprised 277 sequences corresponding to 23 distinct haplotypes. Each color represents a distinct haplotype, and each pie charts represent haplotype composition depending on their frequency in the sampling site. Grey shadow with numbered circle represented the geographical range of marine ecoregion: (1) South Kuroshio, (2) South China Sea Oceanic Island, (3) Sunda Shelf, (4) Southern Java, (5) Lesser Sunda, (6) Sulawesi Sea, (7) Northeast Sulawesi, (8) Papua, (9) Mariana Islands, (10) Maldives, (11) Torres Strait Northern GBR. The grey arrow represents ocean currents and their direction. NEC, North Equator Current; NECC, North Equatorial Counter Current; SEC, South Equatorial Current; EGC, Eastern Gyral Current. All information of detailed names of sampling sites, abbreviation and collection year can be found in Table S2. The map was created from Map by FreeVectorMaps.com (https://freevectormaps.com).

a pattern of isolation-by-distance by performing a Mantel test between genetic ($F_{ST}$) and spatial distances revealed a significant relationship ($r^2 = 0.2144$, $p < 0.05$) for *T. hoshinota* across the IWP (Fig. 4).

# DISCUSSION

The cyanobacteriosponge *T. hoshinota* has become ubiquitous in the Indo-West-Pacific (IWP) region in the past few decades. To understand its distribution, ecology, and evolution, we examined its genetic variation, historical demography, and population differentiation to infer the possible dispersion mechanisms and outbreaks of *T. hoshinota* across the IWP. Our results showed that; (1) *T. hoshinota* is a wide-spread cyanobacteriosponge in the IWP, and both the mitochondrial COI and nuclear ITS2 markers showed a dominant haplotype associated with its dispersion and outbreaks, (2) moderate gene flow existed

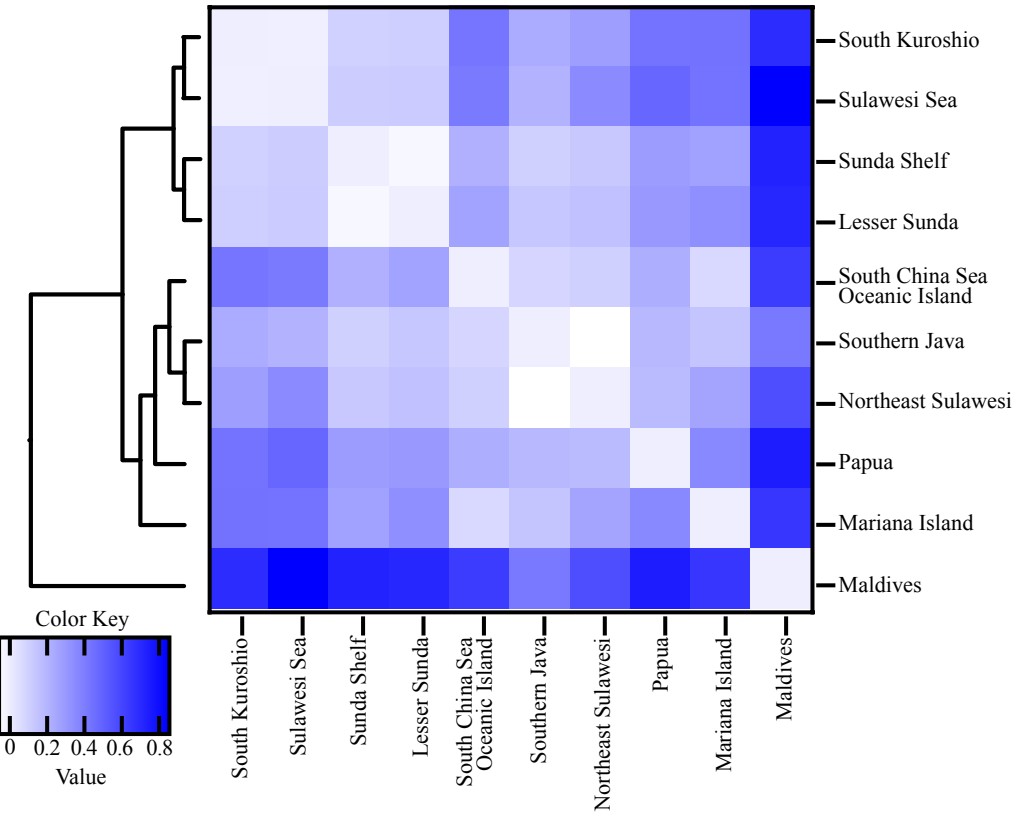

**Figure 3** **Heatmap of pairwise FST values combined with UPGMA clustering between 10 marine ecoregion population of *T. hoshinota*.** Global $F_{ST}$ and average $F_{ST}$ per population following with UPGMA clustering of population were obtained from GraphPad Prism 9. Color key represent the FST values among population comparison. Actual values for each comparison can be found in Table S4.

among most populations in the marine ecoregions of West Pacific, Coral Triangle, and east Indian Ocean, but there was restricted gene flow between the central Indian Ocean and the other three regions, and (3) demographic analyses of most *T. hoshinota* populations were consistent with mutation-drift equilibrium, except for the Sulawesi Sea and Maldives, which had bottlenecks following recent expansions.

## Variability of COI and ITS2 in *Terpios hoshinota*

COI sequences obtained from this study showed extremely low intraspecific variation in *T. hoshinota*. A slow evolution pattern of mitochondrial genomes has been reported in lower metazoans, including certain lineages of poriferans and anthozoans in general (*Shearer et al., 2002*; *Tseng, Wallace & Chen, 2005*; *Wörheide, 2006*; *Huang et al., 2008*; *Chen, Kuo & Chen, 2009*; *Pöppe et al., 2010*; *Sperling et al., 2012*). In the 650-bp COI fragment, only 5 haplotypes with a low *p*- distance of 1.09 and low nucleotide diversity ($\pi = 0.00185$) were identified from 47 *T. hoshinota* specimens collected across the IWP. Slow mitochondrial gene evolution was previously also found in the coralline demosponge *Astrosclera willeyana* across the Indo-Pacific, which contained only three COI haplotypes with a maximum

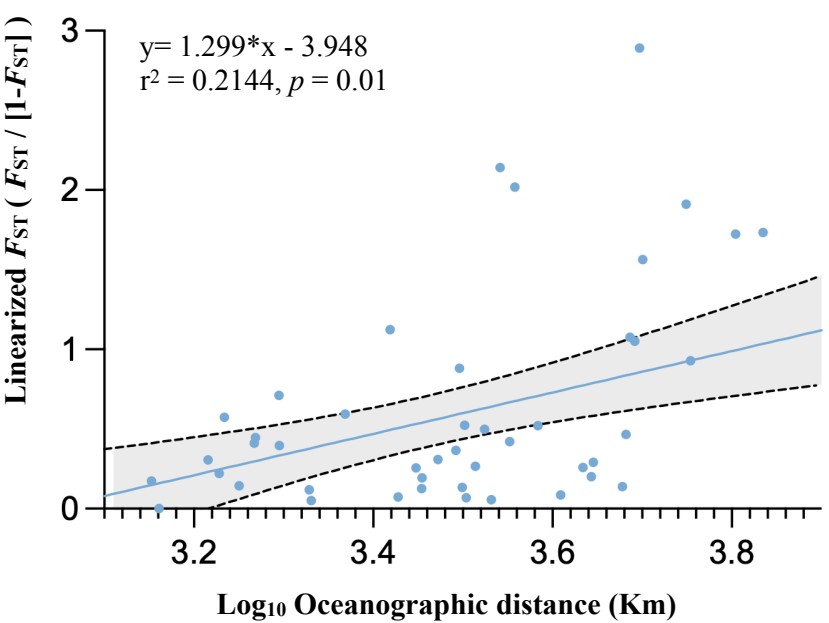

$y = 1.299*x - 3.948$
$r^2 = 0.2144, p = 0.01$

**Figure 4** Graph of relationships between site pairwise linearized FST and log- transformed oceanographic distances (km) of the 11 marine ecoregion population subsets.

p-distance of 0.418 and low nucleotide diversity ($\pi = 0.00049$) (*Wörheide, 2006*). As well, in yellow candle sponges (*Aplysina* spp.) from the Caribbean, there was no interspecific variation in the COI fragment and only 6 nucleotide differences among species observed in the complete mitochondrial genome (19,620-bp) (*Sperling et al., 2012*). Conversely, in recent studies on some deep-sea sponges such as *Poecillastra laminaris,* COI shows significant genetic differences among provinces and geomorphic features in New Zealand (Zeng et al., 2019).

In contrast to COI, ITS2 sequences, although only 128 bp in length, had much higher genetic variation (mean *p*-distance of 1.717 and $\pi$ of 0.0164) and presented 23 haplotypes across the IWP region, suggesting that ITS2 could provide more information for intraspecific inferences in *T. hoshinota.* However, heterogeneous levels of ITS intragenomic polymorphisms (IGPs) in a diverse range of marine sponge taxa have been documented, and it has been recommended that caution must be taken to survey the different levels of IGPs before applying ITS in phylogenetic studies below the species level (*Wörheide, 2006*; *Wörheide, Epp & Macis, 2008*). In this study, we developed ITS2 through a series of sub-cloning evaluations that enabled us to perform PCR and directly sequence the entire region. Relatively low IGPs were detected in ITS2. Following the procedure described in *Fontaneto, Flot & Tang (2015)*, ITS2 could be used as a potential marker to infer the historical demography and population genetics of *T. hoshinota* across the IWP.

## Potential dispersion and outbreak mechanisms: long-range dispersal, local outbreaks, or both?

COI and ITS2 produced similar statistical parsimony networks (SPN) showing a dominant haplotype comprised of several mutational steps evident among sequences from specimens. Although the number of specimens and sampling locations were not the same for the two markers, our results indicated that, by count, ITS2 ($\pi = 0.0164$) evolved 10 times faster than COI ($\pi = 0.00185$). Both markers resolved most of the representative haplotypes from the distributional range of *Terpios hoshinota* in the IWP, finding 5 COI haplotypes out of 47 samples (10%) and 23 ITS2 haplotypes out of 234 samples (9%). In the COI SPN, haplotype C2 was dominant, corresponding well with the most dominant ITS2 haplotype H3 from Southern Kuroshio, H2 from Mariana Islands, H6 from Maldives, and H22 and H23 from Xisha, South China Sea Islands. Interestingly, C2 also corresponded to the COI haplotype previously identified from the Spermonde Archipelago, Indonesia (*vander Ent, Hoeksema & de Voogd, 2016*), and the Kimberley inshore reef, Western Australia (*Fromont, Richards & Wilson, 2019*), suggesting that C2/H3 is the major haplotype in shaping the demography, population genetic structure, and probably outbreaks of *T. hoshinota* in the IWP.

Since its first discovery in Guam (*Bryan, 1973*), occurrences and outbreaks of *Terpios hoshinota* have been reported along the South Kuroshio marine ecoregion towards the South China Sea Oceanic Islands, Torres Strait Northern GBR, marine ecoregions in the Coral Triangle including those of Malaysia and Indonesia, and more recently into the Indian Ocean (Table 1, Table S2). By superimposing the main ocean currents onto the sampling sites and ecoregions (Fig. 2), long-range dispersal, or the "recent expansion, perhaps from Guam" hypothesis could, at least in part, explain how *T. hoshinota* might have dispersed in the IWP. In this hypothesis, *T. hoshinota* is represented by its two most "common" ITS2 haplotypes, H3 and H2, which spread from Guam along the North Equatorial Current (NEC) to the west and the Kuroshio Current to the north to invade reefs in Japan and Taiwan (*Liao et al., 2007*; *Chen, Kuo & Chen, 2009*; reviewed in *Reimer et al., 2011*). These two ITS2 haplotypes also invaded the South China Sea Oceanic Islands along a branch of the Kuroshio Current, entering the South China Sea through the Luzon Strait and then traveling clockwise on the South China Sea Surface Current (SCSSC) to cause outbreaks in Itu Aba, Spratlys *Yang et al., 2018*, and the Sunda Shelf in the Gulf of Thailand (Fig. 2). On the other hand, the other branch of the NEC, the Mindanao Current, turns southward and transported *T. hoshinota* to the reefs around the Coral Triangle. The Mindanao Current also accounts for the trans-equatorial transportation of *T. hoshinota* from the Coral Triangle to invade Kimberley inshore reefs, Western Australia (sharing the C2/H3 haplotype) (*Fromont, Richards & Wilson, 2019*). Trans-equatorial transportation of *T. hoshinota* is further supported by long-term benthic surveys and examination of museum collections, which indicate that *T. hoshinota* may have arrived recently in the Kimberley, possibly after 2014 (*Fromont, Richards & Wilson, 2019*). Similar observations were also seen with regards to the *T. hoshinota* outbreaks at Green Island reefs that appeared in 2007 but were not recorded in ReefCheck surveys between 1999 and 2006 (*Liao et al., 2007*; *Chen, Kuo & Chen, 2009*).
Results from pairwise $F_{ST}$ and isolation-by-distance (IBD) is consistent with the "recent expansion, perhaps from Guam" hypothesis. Only 32.73% of pairwise $F_{ST}$ values were significant, suggesting that there were recent expansion events or moderate gene flow existed among marine ecoregions in the West Pacific, Coral Triangle, and eastern Indian Ocean, but not the central Indian Ocean. The $F_{ST}$ heatmap clearly showed that the Maldives had much higher $F_{ST}$ values compared to counterparts in other marine ecoregions (see Fig. 3 and Table S4). In addition, IBD, a pattern of population differentiation based on the stepping-stone model of population structure in which genetic differences between populations increase with geographic scale, was also significant. Trans-oceanic genetic divergence across the Pacific and Indian Oceans has been documented in many tropical marine taxa (*Crandall et al., 2008*; *Wörheide, Epp & Macis, 2008*; *Bowen et al., 2013*; *Ma et al., 2018*). A phylogeographic study on the lemon-yellow calcareous sponge, *Leucetta chagosensis*, showed that while populations from the Philippines, the Red Sea, the Maldives, Japan, Samoa, and Polynesia were reciprocally monophyletic, suggesting long-standing isolation, the populations along the South Equatorial Current in the south-western Pacific, on the contrary, showed isolation-by-distance effects. It was concluded that the dispersal pattern of *L. chagosensis* was stepping-stone dispersal with some putative long-distance exchanges (*Wörheide, Epp & Macis, 2008*).

Despite the possibility of ocean currents mediating the dispersal of *Teripos hoshinota* in the IWP, the alternative hypothesis of "local outbreak" should also be considered, wherein the sponge may appear in a new environment through occasional or rare events. Once *T. hoshinota* arrives on a new reef, it might remain undetected at low abundances for a long period of time until an environmental disturbance triggers a new outbreak. Larval dispersal of *T. hoshinota* is short, and its pseudoblastula larvae have been shown to crawl out from the oscula and settle rapidly within 1 day, subsequently initiating metamorphosis on substrates consisting of dead coral skeletons and a glass petri dish but not on living corals (*Wang et al., 2012a*; *Wang et al., 2012b*; *Hsu, Wang & Chen, 2013*; *Nozawa, Huang & Hirose, 2016*). The absence of planktonic larvae and having quick settlement runs against basic assumptions of the long-range dispersal hypothesis. However, rafting with floating materials (*e.g.*, pumice), as seen in other benthic organisms, is consistent with long-range dispersal (*Jokiel, 1984*; *Jokiel, 1990*; *Bryan et al., 2012*), and rapid movement caused by typhoons (*Nozawa, Huang & Hirose, 2016*) could be another alternative way for *T. hoshinota* to travel long distances with ocean currents.

Moderate gene flow with IBD and a demographic history of mutation-drift equilibrium without significant expansion supports this hypothesis with respect to *T. hoshinota* dispersal and outbreaks in the IWP. All the indices (Tajima's D, Fu's $F_S$ statistics, mismatch modelling) used to examine the demographic history of *T. hoshinota* based on marine ecoregions showed that most populations were consistent with the model of drift–mutation equilibrium with no evidence of selection, except for populations in the Sulawesi Sea and the Maldives (significantly negative Tajima D, Fu's $Fs$), and skewed unimodal mismatch distribution may suggest sudden population expansions with bottlenecks in these populations. However, Ramos-Onsins' $R_2$, which is more powerful for detecting past demographic events, especially in populations with small sample sizes, remained

non-significant in all populations, suggesting that *T. hoshinota* might have existed in the sampled reefs for a long period of time in relatively low abundances, followed by local outbreaks through asexual propagation and also consequently noted during reef surveys. Previous studies have shown that *T. hoshinota* can increase its coverage through asexual propagation at 1.5–2 times the speed of other sponges ((*Rützler & Muzik, 1993*), Rossi et al., 2015), and spread twice as fast as scleractinian corals (*Elliott et al., 2016a*; *Elliott et al., 2016b*). Therefore, local outbreaks such as those in Japan and Taiwan might have resulted from asexual propagation of the dominant haplotypes within a short period of time, leading to low genetic diversity without any significance in demographic indices.

## Limitation of conventional molecular markers

Contrasting interpretation showing a moderate gene flow detected among the marine ecoregions with a clear pattern of isolation-by-distance (IBD) and demography analyses hinting that *T. hoshinota* populations were in the status of mutation-drift equilibrium without significant expansion might be due to the characteristics of molecular markers. While mtCOI is slow-evolving, ITS2 is too short. Also, both contain few phylogenetic informative sites that provided limited information for genetic analyses. *Terpios hoshinota* is a cyanobacteriosponge containing relatively high density of symbiotic cyanobacteria (*Hirose & Murakami, 2011*; *Tang et al., 2011*) that retards the development of *Terpio*s-specific gene loci. Future applications of the single-cell capture method (*Tang et al., 2009*) and next generation sequencing are expected to obtain in-depth genomic information for genetic research in *T. hoshinota*.

## Population genetic implications for anthropogenic and natural disturbances causing *T. hoshinota* outbreaks

Occurrences of *Terpios hoshinota* have been reported from the West Pacific from the early 1970s (*Bryan, 1973*; *Plucer-Rosario, 1987*), and now extend into the Indian Ocean (*Elliott et al., 2016a*; *Elliott et al., 2016b*; *Montano et al., 2014*; *Fromont, Richards & Wilson, 2019*). Nevertheless, the ecological mechanisms of *T. hoshinota* outbreaks remain uncertain. The outbreaks are suspected to be related to large-scale disturbances caused by coastal development (*Rützler & Muzik, 1993*; *Chen, Kuo & Chen, 2009*) or volcanic eruptions (*Schils, 2012*), which could have resulted in increased turbidity, inshore nutrient run-off, and trace metals such as iron from volcanic ash being present in seawater. These inputs accelerate the propagation of symbiotic cyanobacteria and growth of sponge hosts, resulting in the spread of *T. hoshinota*. Once a population outbreak is established, it can occupy reefs for decades, only disappearing possibly due to strong waves caused by storms, but still having the ability to revert to outbreak proportions at a later time (*Reimer et al., 2011*; *Schils, 2012*; *Yomogida et al., 2017*). As such, occurrence and outbreak through "long-distance dispersal" and "local outbreak" are non-mutually exclusive. Meaning both of these two mechanisms may have resulted in the outbreaks of *T. hoshinota* in the past 20 years. It is important to continue long-term ecological studies using newly developed genomic applications to unveil the outbreak mechanisms of *Terpios hoshinota*, particularly given the increasing impacts of climate change.

## CONCLUSION

This study illustrates the possible dispersion mechanisms and outbreak of *T. hoshinota* among the IWP. Our results, based on genetic variation, historical demography, and population differentiation suggest that the potential outbreak mechanisms of *T. hoshinota* may result of both long-range dispersal and local outbreaks. However, we advise caution due to insufficient genetic variation and the lack of *Terpios*-specific gene loci on conventional molecular marker. Further investigation through genomic investigation might be able to unravel more comprehensive information about this sponge.

### Funding

This work was supported by grants from the National Science Council (NSC 101-2621-B-127–001, MOST 103-2311-B-127-001) to Jih-Terng Wang. Funding for Keryea Soong was provided by MOST 98-2611-M-110-003-MY3. Hawis Madduppa was funded by the Indonesian Ministry of Research, Technology & Higher Education (KLN RISTEKDIKTI scheme, 2016-2018) No. 011/SP2H/LT/DRPM/IV/2017. Chun Hong Tan was supported by a project fund from Department of Fisheries Malaysia (formerly Department of Marine Park Malaysia), number: P23 17100 005001. Chaolun Allen Chen was funded by Academia Sinica Thematic Grants and MOST Grants between 2010-2016. The funders had no role in study design, data collection and analysis, decision to publish, or preparation of the manuscript.

### Grant Disclosures

The following grant information was disclosed by the authors:
National Science Council: NSC 101-2621-B-127–001, MOST 103-2311-B-127-001, MOST 98-2611-M-110-003-MY3.
Indonesian Ministry of Research, Technology & Higher Education (KLN RISTEKDIKTI scheme, 2016-2018): 011/SP2H/LT/DRPM/IV/2017.
Department of Fisheries Malaysia: P23 17100 005001.
Academia Sinica Thematic Grants and MOST.

### Competing Interests

James Reimer is an Academic Editor for PeerJ. Hock Chark Liew is employed by Alchemy Laboratory & Services.

### Author Contributions

- Savanna Wenhua Chow conceived and designed the experiments, performed the experiments, analyzed the data, prepared figures and/or tables, authored or reviewed drafts of the article, and approved the final draft.
- Shashank Keshavmurthy performed the experiments, analyzed the data, prepared figures and/or tables, authored or reviewed drafts of the article, and approved the final draft.

- James Davis Reimer analyzed the data, authored or reviewed drafts of the article, and approved the final draft.
- Nicole de Voogd analyzed the data, authored or reviewed drafts of the article, and approved the final draft.
- Hui Huang analyzed the data, authored or reviewed drafts of the article, sample collection, and approved the final draft.
- Jih-Terng Wang analyzed the data, authored or reviewed drafts of the article, sample collection, and approved the final draft.
- Sen-Lin Tang analyzed the data, authored or reviewed drafts of the article, sample collection, and approved the final draft.
- Peter J. Schupp analyzed the data, authored or reviewed drafts of the article, and approved the final draft.
- Chun Hong Tan analyzed the data, authored or reviewed drafts of the article, sample collection, and approved the final draft.
- Hock-Chark Liew analyzed the data, authored or reviewed drafts of the article, sample collection, and approved the final draft.
- Keryea Soong conceived and designed the experiments, analyzed the data, authored or reviewed drafts of the article, and approved the final draft.
- Beginer Subhan analyzed the data, authored or reviewed drafts of the article, sample collection, and approved the final draft.
- Hawis Madduppa analyzed the data, authored or reviewed drafts of the article, sample collection, and approved the final draft.
- Chaolun Allen Chen conceived and designed the experiments, analyzed the data, authored or reviewed drafts of the article, and approved the final draft.

## Field Study Permissions

The following information was supplied relating to field study approvals ({i.e.}, approving body and any reference numbers):

For Guam, samples were collected with permission of the Department of Agriculture Division of Aquatic and Wildlife Resources (DAWR) Guam and certificate of origin from DAWR was obtained prior to shipping.

Permit is not required to collect sponges in Green Island and in China.

For Indonesia permit was issued by the Indonesian Ministry of Research, Technology & Higher Education (KLN RISTEKDIKTI scheme, 2016-2018) No. 011/SP2H/LT/DRPM/IV/2017.

For Malaysia, sampling permit was issued by the Marine Park and Resource Management Division, Department of Fisheries Malaysia (formerly Department of Marine Park Malaysia) with a Sampling permit: JTLM 630-7 JLD. 4(8)

Samples from Australia were part of Census of Marine Life / Census of Coral Reefs, with permit No. G32313.1 issued by the Great Barrier Reef Marine Park Authority.

## Data Availability

Raw data in this work is Sequences of the sponge - ITS2 and COI markers.

All COI sequences obtained in this study were submitted to GenBank with accession numbers MZ914515–MZ914532 and OK576574–OK576593.

All ITS2 sequences were submitted to GenBank with accession numbers MZ468904–MZ469137.

## Supplemental Information

Supplemental information for this article can be found online at http://dx.doi.org/10.7717/peerj.13451#supplemental-information.

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
