# Peer review of "Population genetics and demography of the coral-killing cyanobacteriosponge, Terpios hoshinota, in the Indo-West Pacific"

_PeerJ, doi:10.7717/peerj.13451_

## Round 0.1 · original submission · Major Revisions

I now have three substantial reviews from experts in the general area of invertebrate population genetics, and while the reviewers are in agreement over the value of the data and topic of investigation, they have concerns that will require major revisions and a second review. In particular, see the comments on statistical analyses provided by Reviewer 1 (with attached PDF). Please provide a point-by-point response to each of the reviewers' comments with your revised manuscript.

·

Basic reporting

Needs Genbank numbers for genetic data; figures and tables need a little improvement; a couple of citations to add, as detailed below.

Experimental design

I have some suggestions about statistics in the attached pdf, but I think they are fixable.

Validity of the findings

I commend the authors on their hard work. I'm sure that assembling this dataset was an enormous effort! Nice job! There is lots to like in this paper, and I think that it could be an important contribution.

However, to make sure the author's hard work is not wasted, this paper needs major changes to the analysis and how the results are discussed. First of all, the analysis of genetic variation needs to better take sample size into account. As shown below, doing so will change their conclusions significantly. Second, the authors need to let the data "speak". Instead, they are trying to find a way to force the data to support their a priori hypothesis: that the species has spread from Guam to other locations recently. A more balanced analysis of their data is unlikely to find support for this hypothesis, and I explain why below. There is no point in collecting data if you will say that it supports your a priori conclusion no matter what it looks like.

Please see the details in the attached pdf for more info.

Additional comments

Please see the attached pdf

Reviewer 2 ·

Basic reporting

This study investigate the genetic variation of coral -killing Cyanobacteriosponge Terpion hoshinota, which is distributed widely in the Indo-west pacific. There results suggest H5 are most abundant haplotype in Indo-West pacific and H12, which originated from Guam, has invaded other regions with slight genetic change. It was found that some genotype contribute out-breaks of Terpios. Authors sampled a large number of Terpios over a wide area and the results showing the spread of certain genotypes were very interesting. One thing that should be pointed out throughout is that there should be some prior research on genetic variation in terpions, but this is not mentioned. Mentioning previous studies in DISCUSSION and INTRODUCTION, and discussing the differences between their results and the results here, is necessary to objectively evaluate the present data.

Experimental design

Another point of concern is the Terpios specificity of the primers used for genetic variation. The authors first used a universal primer for the ITS2 region (ITS4 and EF3RCNL), but they need to explain why they used this universal primer. Isn't this a primer used for fungal and plant diversity? Or is this a commonly used primer for this type of sponge? Please add more explanation. In the results, there should be a confirmation that the ITS region sequence used for genetic variation is the sequence from Terpios. For example, it is necessary to check the similarity of the ITS of the closely related Songe species. I think authors need to refer about sampling permits for all sampling sites (e.g. Great Barrier Reef). Please confirm.

Validity of the findings

Please provide the accession numbers in the Gene Bank for the COI gene sequence and the ITS gene sequence that you have obtained. Does the raw data include the ITS region sequence? There is only one GBR data in Table 1, but there are two in the row data. Why is that? In raw data, some of the file names use Abbrevation from Table 1, and some do not. It is difficult to recognize, so please make it consistent.

Figure S1
Please exchange to a file with better resolution.Is this figure really necessary, and how does it differ from Figure 2? If so, why don't you add more explanation about this figure?

Additional comments

Minor comments
L.55, IWP region is Indo-West pacific region?
Please define it
L. 58, The results of this study indicate・・
"suggest" is more appropriate than "indicate.
L. 136, Please add references about genetic diversity in T. Hoshinota.
L. 144, Please mention the permission of great barrier reef
L.175, Please mention sequencer machine name, and more details of sequencing method. Which kit did you use? Did it performed with Sangar sequencing method?
L186, rDNA sequences of specimens obtained from the GenBank database.
Please insert the Accession No. of the rDNA sequences
L. 211-212, Only Guam is not an abbreviation. Please unify.
L. 217~ , For the ITS region, explain the sequence similarities between several different sponge taxa.
L.240, Please change "Xisha" to an abbreviation.
   I cannot recognize the name
L. 261, "Hap12" or "H12"?
Use the same haplotype abbreviation.
L. 266, Hap 1 is shown in Fig. 3? Please add Hap 1 in Fig. 3.
L. 290-293, Please add reference.

Reviewer 3 ·

Basic reporting

It will be good to get one or two native English speakers to proofread the manuscript to improve the language.

Sufficient references. Context provided.

Minor improvements required for figures, suggestions in attached word file.

Self-contained with relevant results.

Experimental design

Original primary research.

Research question is fine and knowledge gap is identified.

Investigation performed to standard.

Minor improvements required Methods, suggestions in attached word file.

Validity of the findings

Results are novel.

Conclusions can be improved after issues are addressed.

Annotated reviews are not available for download in order to protect the identity of reviewers who chose to remain anonymous.

---

## Round 0.2 · Major Revisions

While the Reviewers note that this manuscript has improved, they indicate that it is not ready for publication. Most importantly, Reviewer 2 still has basic questions about the sequencing methodology and Reviewer 3 requests clarification regarding the study's hypothesis and conclusions. The manuscript will need to be returned to these reviewers for further consideration.

·

Basic reporting

Sufficient, minor revisions suggested below

Experimental design

Sufficient, minor revisions suggested below

Validity of the findings

Sufficient, minor revisions suggested below

Additional comments

Sufficient, minor revisions suggested below

Reviewer 2 ·

Basic reporting

Terpios habitat on coral and sometimes threaten the survival of coral reefs. Their genetic differentiation and regional differences are interesting, that are suggesting how Terpios began to spread on coral reefs. I felt a big change from the previous version. However, what makes me suspicious in this paper is that it is still difficult to determine whether the ITS2 sequence is actually derived from Terpios based on the information provided. This is because, all Terpios ITS2 sequences provided here, are homologous to the Terpios sequences in this paper, and other ITS sequence of sponges were not detected by homology search,by NCBI blastn.

Experimental design

I feel anxious about universal primers (ITS4, EF3RCHL) that are used for the identification of the ITS2 region of Terpins. This primer set is a universal primer for fungi?, and Terpios is not fungi. It might have been better to design primers based on the known sponge nucleus sequence. Anyway, there are still doubts about the method for identification of ITS2 sequence and the information of ITS2 sequence.
The author should provide the sequence information obtained by the first PCR using universal primers (ITS4, EF3RCHL), and prove that it is actually derived from Terpios by alignment analysis or by phylogenetic analysis.

Validity of the findings

'no comment'

Additional comments

'no comment'

Annotated reviews are not available for download in order to protect the identity of reviewers who chose to remain anonymous.

Reviewer 3 ·

Basic reporting

No comment

Experimental design

No comment

Validity of the findings

Objectives and conclusions are not well stated.

Objectives
1) To sample T. hoshinota from its currently known distribution in the IWP and examine the variation in its COI and nuclear ribosomal internal transcribe spacer (ITS) sequences.

All its currently known distribution in the IWP from existing literatures? If not, please list the localities sampled.

Achieved

2) To elucidate the population genetic structure and demographic history of T. hoshinota
Achieved

3) To propose hypotheses on how T. hoshinota dispersal and colonization in the IWP?
I have difficulties identifying and understand the hypothesis on T. hoshinota dispersal and colonization in the IWP in this study. Please state clearly and elaborate if there's a hypothesis.

Is “potential outbreak mechanisms of T. hoshinota may result of both long-range dispersal and local outbreaks” in the Conclusion the hypothesis that your study aims to obtain? If yes, please elaborate and distinguish how your hypothesis is different from the typical sponge dispersal and typical local outbreak.

Conclusion
Please elaborate the how the outbreak mechanism in your study is different from typical outbreak mechanisms in your Discussion. An example of a typical outbreak is when environmental changes favour expansion of T. hoshinota population that is already established and detrimental to health of coral reefs.

Please elaborate and distinguish how your "long-range dispersal" is different from the typical sponge dispersal, and how is it relevant as your proposed hypothesis.

Please discuss the possibility of T. hoshinota causing an outbreak “immediately” or in a season after long-range dispersal to support your hypothesis. Are there studies or data on a population of T. hoshinota establishing on a reef after long-range dispersal and causing an outbreak immediately or in the same season or short period of time?

Additional comments

No comment

---

## Round 0.3 · accepted · Accept

All three reviewers have found the manuscript improved and recommend acceptance. Please see the comments of Reviewer 2 as you make final edits to the manuscript before publication.

·

Basic reporting

I looked through the authors responses to my minor comments from round 2, and am fully satisfied

Experimental design

I looked through the authors responses to my minor comments from round 2, and am fully satisfied

Validity of the findings

I looked through the authors responses to my minor comments from round 2, and am fully satisfied

Reviewer 2 ·

Basic reporting

The unclear point about the Terpios sequence is improved in the revised ver. There are some points where the reply comment is not reflected in the text, so I decide it is a minor revision, but it's OK that it is almost "accept".

Experimental design

no comment

Validity of the findings

no comment

Additional comments

Before, accept, please check the following points.

Please confirm Figure legend (I didn't find).
The figures, those are about "This is the result of the array alignment. This figure is included in the supplementary section"were also not included in supplementary section?
.

Reviewer 3 ·

Basic reporting

I'm satisfied with the authors' revisions.

Experimental design

I'm satisfied with the authors' revisions.

Validity of the findings

I'm satisfied with the authors' revisions

Additional comments

I'm satisfied with the authors' revisions and I recommend the manuscript for publication.